# Population genomics identifies a distinct *Plasmodium vivax* population on the China-Myanmar border of Southeast Asia

**Awtum M. Brashear**[1,2], **Qi Fan**[3], **Yubing Hu**[4], **Yuling Li**[4], **Yan Zhao**[4], **Zenglei Wang**[5], **Yaming Cao**[4], **Jun Miao**[1], **Alyssa Barry**[6], **Liwang Cui**[1] *

**1** Department of Internal Medicine, University of South Florida, Tampa, Florida, United States of America, **2** Pennsylvania State University, University Park, Pennsylvania, United States of America, **3** Dalian Institute of Technology, Dalian, Liaoning Province, China, **4** Department of Immunology, College of Basic Medical Sciences, China Medical University, Shenyang, Liaoning, China, **5** MHC Key Laboratory of Systems Biology of Pathogens, Institute of Pathogen Biology, Chinese Academy of Medical Sciences and Beijing Union Medical College, Beijing, China, **6** Infection Systems Epidemiology, School of Medicine, Faculty of Health, Deakin University, Geelong, VIC, Australia

* liwangcui@usf.edu

**Data Availability Statement:** Sequencing results from this study are publicly available in the NCBI Sequence Read Archive within BioProject PRJNA603279.

## Abstract

*Plasmodium vivax* has become the predominant malaria parasite and a major challenge for malaria elimination in the Greater Mekong Subregion (GMS). Yet, our knowledge about the evolution of *P. vivax* populations in the GMS is fragmental. We performed whole genome sequencing on 23 *P. vivax* samples from the China-Myanmar border (CMB) and used 21 high-coverage samples to compare to over 200 samples from the rest of the GMS. Using genome-wide single nucleotide polymorphisms (SNPs), we analyzed population differentiation, genetic structure, migration and potential selection using an array of methods. The CMB parasites displayed a higher proportion of monoclonal infections, and 52% shared over 90% of their genomes in identity-by-descent segments with at least one other sample from the CMB, suggesting preferential expansion of certain parasite strains in this region, likely resulting from the *P. vivax* outbreaks occurring during this study period. Principal component, admixture, fixation index and phylogenetic analyses all identified that parasites from the CMB were genetically distinct from parasites from eastern parts of the GMS (Cambodia, Laos, Vietnam, and Thailand), whereas the eastern GMS parasite populations were largely undifferentiated. Such a genetic differentiation pattern of the *P. vivax* populations from the GMS parasite was largely explainable through geographic distance. Using the genome-wide SNPs, we narrowed down to a set of 36 SNPs for differentiating parasites from different areas of the GMS. Genome-wide scans to determine selection in the genome with two statistical methods identified genes potentially under drug selection, including genes associated with antifolate resistance and genes linked to chloroquine resistance in *Plasmodium falciparum*.

**Funding:** LC received funding from the National Institute for Allergy and Infectious Diseases, The National Institute of Health (NIH) (U19 AI089672). AB received funding from the National Library of Medicine (NIH) (T32 LM012415). The funders had no role in study design, data collection and analysis, decision to publish, or preparation of the manuscript.

**Competing interests:** The authors have declared that no competing interests exist.

## Author summary

*Plasmodium vivax* is an understudied malaria parasite compared to *P. falciparum* despite that it is the most common *Plasmodium* species outside of Africa. In the Greater Mekong Subregion (GMS), the increased proportion of *P. vivax* proves its resilience to conventional malaria control measures. Within the GMS malaria incidence is highly heterogeneous, typified by more intensive malaria transmission along international borders. Understanding the transmission between countries and tracking parasite introduction are therefore essential to eliminating malaria within this region. The China-Myanmar border (CMB) presents such an example wherein China has eliminated autochthonous malaria cases, while Myanmar has high malaria incidence. Malaria on the CMB is nearly entirely due to *P. vivax*, yet few studies investigated the genetics and evolution of the *P. vivax* populations in the area. Here we used whole-genome sequencing for a holistic analysis of *P. vivax* from the CMB and compared them to those from other sites of the GMS. Parasites on the CMB had a significantly higher proportion (75%) of monoclonal infection than parasites from other regions. Many of the CMB parasites showed significant genetic sharing that is consistent with the result of clonal expansion, consistent with the malaria outbreak occurring during the study period. While *P. vivax* parasites from the entire GMS were substantially mixed with no evidence of significant gene flow barriers, those from the CMB were more genetically distinct from other populations. Genome-wide scans for selection identified genes potentially under selection, and especially notable are genes associated with sulfadoxine/pyrimethamine resistance. Genes also under selection include those potentially encoding membrane channels and transporters, which were associated with drug resistance in *P. falciparum*. Moreover, this population genomic study also identified a set of 36 single nucleotide polymorphisms, which could serve as a barcode for differentiating parasites from various regions of the GMS, a task that is important for the final phase of regional malaria elimination.

## Introduction

The Greater Mekong Subregion (GMS) in Southeast Asia is in the pursuit of malaria elimination, aiming to achieve this goal by 2030 [1]. The GMS consists of six countries, Laos, Vietnam, Myanmar, Thailand, Cambodia, and Yunnan and Guangxi provinces of China, and is especially concerning due to the repeated emergence of drug resistance in *Plasmodium falciparum* to frontline treatments [1–5]. Within the GMS, malaria distribution is not uniform with border regions having much higher levels of malaria transmission while central regions of the countries are mostly malaria-free [6–9]. Myanmar has especially high country-wide malaria endemicity, and malaria re-introduction into the neighboring countries, such as China and Thailand, is a major concern [9–11]. Importantly, *Plasmodium vivax* is the dominant species of malaria in Myanmar and on the China-Myanmar border (CMB). On the CMB, *P. vivax* was the cause of multiple malaria outbreaks in the past decade, including one in 2013 and one in 2016 [7]. It is unclear exactly what caused these outbreaks, but some potential causes include increased drug resistance or the expansion of introduced or relapsed parasite lineages [7]. It is crucial that we monitor the epidemiology of *P. vivax* along the CMB in order to understand the composition of the local population and its relationship with nearby populations, allowing us to implement targeted control efforts.

To track the progress of malaria elimination in the GMS, population genetic studies have been undertaken using polymorphic antigen and microsatellite markers [8–10,12,13]. These

studies revealed population differentiation among endemic areas, and identified potential routes of parasite introduction across borders. Given the potential shared ancestry of parasite populations within this region in the recent past, the use of whole genome sequencing (WGS) technology offers an unprecedented chance of identifying parasite migration within relatively small regions, and characterizing evolutionary trajectories of parasite populations and selection pressures in the *Plasmodium* genomes [14–18]. While some studies have focused on dissecting parasite population structures within individual countries [15,18], others aimed to present a global view of the parasite populations [14,16]. However, a more comprehensive analysis of the *P. vivax* populations in the GMS would allow us to home in on regional characteristics to identify signals of selection, potential inter-country parasite movement, and genetic structures as well as to identify a set of genetic barcode for differentiating parasites at a smaller geographic scale in the GMS. Further, while *P. falciparum* parasites from the CMB have distinct ancestry from those from elsewhere in the GMS [19,20], few studies have compared *P. vivax* from the CMB to other parts of the GMS. Here, we have procured clinical *P. vivax* parasite isolates from the CMB and performed WGS. We characterized parasites in the region in terms of clonality and genetic diversity. Additionally, by comparing the new genomes to those collected elsewhere in the GMS we were able to make inferences about evolution and transmission of *P. vivax*, providing guidance on regional control activities targeted at this resilient parasite.

## Methods

### Ethics statement and sample collection

This research was approved by local Bureau of Health of Kachin and the Institutional Review Board of Pennsylvania State University IRB #34319. Twenty-three patients with acute *P. vivax* malaria presented at the hospital and clinics of Laiza Township, Kachin State, Myanmar and Nabang Township, Yunnan Province, China were recruited in 2013 after obtaining written informed consent. Malaria diagnosis was performed by microscopic examination of Giemsa-stained blood smears. Five milliliters of venous blood were drawn by venipuncture from each patient and filtered to remove human leukocytes as previously described [21]. Parasites were released after lysis of the red blood cells (RBCs) with saponin, pelleted by centrifugation, and stored at -80˚C until DNA extraction.

### DNA extraction and library preparation

Parasite DNA was extracted from the parasite pellets using the QIAamp DNA Mini Kit (Qiagen, Hilde, Germany) and air-dried for storage. DNA from each sample was resuspended in 20 μL of water and DNA concentration was measured on a Qubit Fluorometer. DNA libraries were prepared using the TruSeq Nano DNA Library Prep Kit with up to 100 ng of DNA, which was sheared to create 350 nt inserts. DNA libraries were sequenced on the HiSeq 2500 in rapid mode to create 150 bp paired-end reads. DNA reads were trimmed for quality, removing areas with a Phred score under 20 using trimmomatic [22].

### Publicly available sequences

Fastq files for samples from other parts of the GMS were obtained from publicly accessible databases. These include 150 samples from the MalariaGen *P. vivax* genome data release (ENA accession numbers available at malariagen.net/data/p-vivax-genome-variation-may-2016-data-release) [16], 28 samples from NCBI bioprojects PRJNA240356–PRJNA240533 [14], 37 samples from NCBI bioproject PRJNA420510 [18], 78 samples from NCBI bioproject PRJNA295233 [15], and 6 samples from NCBI bioproject PRJNA284437 [23]. These sequences

were processed similarly to the 23 new samples, though Madagascar samples were single-ended and therefore aligned under the single-ended algorithm.

## Alignment to reference and variant calling

Reads for all samples underwent identical read trimming for quality using trimmomatic to trim region with a phred score < 20 [22]. Reads were then aligned to the human genome hg38 to account for human contaminating reads. Surviving reads were aligned to the *P. vivax* P01 reference (PlasmoDB release 35) using BWA MEM [24]. Depth for each base was reported using samtools [25] and sample depths were reported as averages across all bases, including those with zero coverage. GATK v 4.1 Haplotype Caller in GVCF mode was used for variant calling [26].

## Variant filtering and sample selection

Low-quality single nucleotide polymorphisms (SNPs) were removed using VCFtools v 0.1 [27]. We removed variants with a minor allele frequency less than 0.01, quality score < 40, and variants which were not biallelic. To eliminate highly variable regions which could result in poor mapping quality and mis-called variants, we calculated the coefficient of variation (CV) by dividing the standard deviation over the mean, for coverage of 1000 bp windows surrounding each base pair. Windowed CV values for each base pair were averaged among 12 high-coverage isolates from this data collection. Base pairs in highly variable regions (CV > 1) were removed from the dataset. We also used these calculations to judge subtelomeric variation to remove SNPs from the end of chromosomes. Additional ranges removed due to likely hyper-variability included a section of chromosome 12 which contains *MSP7* genes (792,292–818,496) and a section on chromosome 10 which encodes *MSP3* genes (1,330,765–1,364,791). Samples with 85% or less of all variants called were removed from further analysis, while those remaining were considered high-quality samples. Variants were removed if they were not callable in 95% or more of the high-quality samples. We used MoiMix to determine the $F_{ws}$ metric, which was used for inbreeding analysis as well as to inform clonality. Samples which had an $F_{ws}$ over 0.95 were considered as monoclonal.

## Functional annotation of SNPs

We used snpEff to assign functional annotation to each SNP in our dataset [28]. VCFtools was used for extracting allele frequencies for subpopulations. In the case of *P. vivax* dihydrofolate reductase (*pvdhfr*) F57I/L, we created a custom script to extract samples which had amino acid changes at either or both of position 1077530 or 1077532 which were classified as alternative I/L genotypes.

## $F_{ST}$ analyses, regional barcode and principle component analysis (PCA)

We calculated the pairwise $F_{ST}$ value for each called SNP between each pair of countries with 10 samples or more using Plink [29]. We took an average of all SNPs for the total $F_{ST}$ value and applied bootstrapping to obtain the 97.5% and 2.5% quantiles as our confidence interval. The top 10 SNPs from each of the pairwise calculations were extracted and redundant SNPs were removed to create a collection of 53 high-$F_{ST}$ SNPs called across 265 samples. This SNP set was further refined by removal of SNPs which displayed linkage disequilibrium (LD) to create a SNP barcode for the GMS parasite population.

PCA was performed using the genome-wide SNP set as well as top 53 high-$F_{ST}$ SNPs. For PCA, SNPs with a LD score over 0.2 were removed, which left 45,676 genome-wide SNPs and 36 high- $F_{ST}$ SNPs. Because $F_{ST}$ calculations were performed independently on each SNP at the population level, we kept polyclonal samples to reduce sample size loss. For consistency between $F_{ST}$ and PCA outputs during barcode evaluation, we also performed the PCA with the full set of high-quality samples, including those which may be polyclonal, and then confirmed overall topology with monoclonal samples. We also extracted SNPs from three global barcode publications and checked their variability within our dataset by quantifying called SNPs [30–32].

## Admixture analysis

We used the ADMIXTURE program v1.3.0 for admixture analysis [33]. We selected monoclonal samples and chromosomal SNPs which were pruned for linkage disequilibrium as suggested by the ADMIXTURE manual using Plink (50kb windows sliding by 10, 0.2 LD threshold). To select the optimal number of populations, we ran 5-fold and 10-fold cross validation on models with k values between 1 and 10, choosing that with the lowest cross-validation result. The best models were repeated under 1000 bootstrap replicates to obtain standard error values.

## Phylogenetic analysis

We extracted 41,008 high-quality SNPs present in 163 monoclonal samples and three samples from Madagascar in phylogenetic analysis with a minor allele frequency (MAF) in this subgroup above 0.05. These SNPs were used as input for RAxML-NG [34] to generate 20 starting trees—10 random and 10 parsimony starting trees. Then a maximum-likelihood tree was constructed with 1000 bootstraps using the GTR substitution matrix and Lewis ascertainment bias correction to account for using only variable regions. Trees were plotted using APE [35]. To demonstrate that structure was not biased by removal of low-MAF SNPs, we repeated maximum-likelihood tree creation 20 times with 5,000 random SNPs without the MAF cutoff.

## Effective migration surfaces, identity-by-descent (IBD) and Mantel test

We used the program EEMS to model parasite migration throughout the GMS. We chose our 162 high-quality monoclonal samples from the GMS with 191,326 SNPs. We used 400 demes to which all samples were assigned to the nearest one. We used 20 million Markov chain Monte Carlo iterations, including 10 million burn iterations, and 99,999 thin iterations in order to achieve chain convergence.

We used IsoRelate to identify pairwise genomic regions that are identical by descent within the GMS [36]. IsoRelate is capable of differentiating between monoclonal and polyclonal samples, and therefore we included all 269 high-quality samples and 191,326 high-quality SNPs. Regions of IBD were included if there were 10 SNPs or more within the segment and it was of 25,000 bp or longer. We found the proportion of the entire genome which was occupied by IBD segments within each pairwise comparison of samples, using 0.01 and 0.9 to signal IBD of different degrees. To confirm results with a genetic distance agnostic program, we repeated parts of this analysis with hmmIBD [37], using only monoclonal samples and entering the CMB samples and other GMS samples as two separate populations in accordance with admixture results. The two programs differ primarily by their sensitivity to population structure and clonality. Mantel testing was performed using APE [35], while km distances used for distance matrices were calculated with geosphere v 1.5 (https://cran.r-project.org/package=geosphere) based on latitude and longitude coordinates.

When selecting representative samples for additional confirmation, we chose the highest coverage monoclonal sample from each of the 4 clusters with IBD sharing above 0.9 and eliminated all other clustered samples.

### Scans for selective sweeps

Many statistics for detecting selective sweeps, such as those used here, are based on segments of homozygosity. Because removal of many consecutive SNPs prone to mismapping, as we performed to limit false SNP calls, may cause false positives, and areas prone to mismapping are largely on chromosome ends, we chose additional conservative masking for the end of chromosomes, where areas with high variability coverage were frequent, excluding at least 10% of each end on each chromosome (S3 Table). In some cases, we extended this section to be longer to account for more extended presence of antigenic variable genes on some chromosomes. For nSL analysis we used the 16 monoclonal samples from the CMB as input to Selscan [38]. Results were normalized within 100 allele frequency bins, then transformed into their absolute values before plotting and selecting the top 1% of values. For the analysis of cross-population extended haplotype homozygosity (XP-EHH), we also used Selscan and normalized resulting values by allele frequency, but we did not adjust to absolute value in order to ascertain selection in the reference vs sample population. We also used isoRelate to construct networks of IBD sharing at selected genes. Since there is no accepted genetic map for *Plasmodium vivax* P01, we used LDHat (https://github.com/auton1/LDhat) to estimate recombination rates at each SNP within clonal GMS samples, and extrapolated an estimated genetic map, which was used in IBD and EHH-based analyses.

## Results

### SNP and sample selection

We collected 23 *P. vivax* clinical isolates in 2013 from the CMB at Laiza, Myanmar and Nabang, China (Fig 1A). WGS reads were trimmed for quality then aligned to the reference genome P01. The median coverage for all 23 samples was 29.13 (S1 Table). Additionally, 302 published whole genome sequences from other parts of the GMS were retrieved, while three genomes from Madagascar were included for comparison [14–16,18,23] (S2 Table). After variant calling, SNPs with a MAF under 0.01 and a quality below 40 were removed. Additionally, to minimize the effect of mismapping on repetitive genes, genomic regions with coverage variability above 1 such as the subtelomeric regions were masked (S1 Fig). These masked hypervariable regions include mostly multigene families such as *vir*, *Plasmodium* exported proteins of unknown function, *pvmsp7* and *pvmsp3* genes (S3 Table). After quality control, 259,380 high-quality, biallelic SNPs were retained for population genetic analysis, of which 191,326 were on chromosomes 1–14, which were considered for genomic scans, while the rest were on unassigned scaffolds. For the 259,380 high-quality core SNPs, 102,849 (39.7%) were in coding regions, and 58,145 (22.4%) were predicted to be non-synonymous SNPs (S2 Fig). Samples missing more than 5% of the high-quality core SNPs were removed, leading to the retention of 269 whole genome sequences for population analysis, including 21 new samples from the CMB (Fig 1A). Other GMS samples include 136 from Cambodia (8 from Battambang, 2 from Preah Vihar, 9 from Kampot, 59 from Oddar Meanchey, 22 from Pursat, and 36 from Ratana-kiri), 91 samples from Thailand (3 from Si Sa Ket Province and 88 from Tak Province), 10 samples from Vietnam, five samples from Yunnan Province, China, and two samples from Laos (Fig 1B). These *P. vivax* genome sequences represent collections in 2006–2007 (16

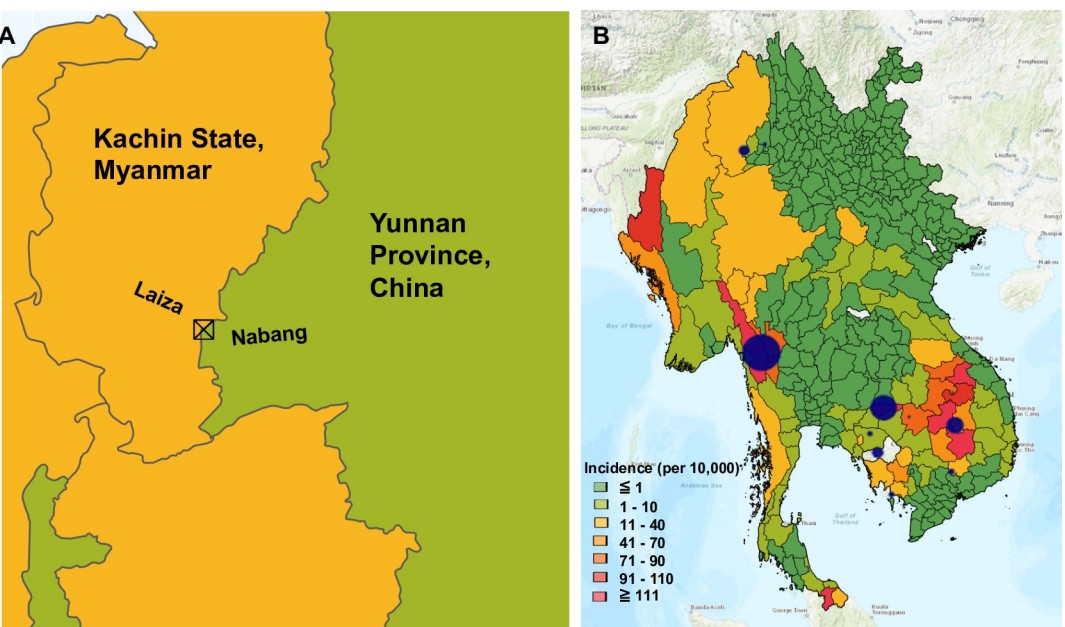

**Fig 1. Sample locations used for this study.** A) Sample collection site for new samples on the China-Myanmar border. B) Sample sites from across the Greater Mekong Subregion. Blue circles represent different locations samples were obtained from, either in this study or in those which were previously available. Bigger circles represent more samples ranging from 2 to 88, and only those which passed initial quality screening were included in analysis.

samples), 2009–2012 (189 samples), 2013 (28 samples), and 2014 (36 samples) (S2 Table) [15–18,39].

## Clonality and inbreeding within the GMS

For the 269 WGS samples, 163 samples had a within sample fixation index $F_{ws}$ of 0.95 or higher, suggestive of monoclonal infections (S2 Table). Among the GMS populations, the CMB samples had the highest median $F_{ws}$ (0.994) (Fig 2A); 16 (76%) had an $F_{ws}$ of $\geq$0.95 (Fig 2B). In comparison, Thai samples had a median $F_{ws}$ of 0.985 and 59/91 (64.8%) high-quality samples had an $F_{ws}$ of $\geq$0.95, followed by Cambodia samples with a median $F_{ws}$ of 0.982 and 77/136 (56.6%) samples having an $F_{ws}$ of $\geq$ 0.95. Pairwise comparison showed that the CMB samples were the only ones that significantly differed from samples from other GMS locations ($P < 0.05$, Wilcoxon rank sum test).

## Population differentiation and structure within the GMS

To determine whether parasite populations in the GMS are genetically differentiated, we first performed pairwise comparison of the $F_{ST}$ statistic among populations with $\geq$10 high-quality samples. This analysis revealed an overall low genetic differentiation of parasite populations within the GMS with $F_{ST}$ ranging from 0.0020 to 0.1044 (Table 1). Whereas the CMB population showed substantial genetic divergence from the rest of the GMS populations ($F_{ST}$ = 0.0891–0.1044), parasite populations from Vietnam, Cambodia and Thailand showed marked genetic similarity ($F_{ST} < 0.02$).

To elucidate the population structure of *P. vivax* within the GMS, PCA showed that parasites from Cambodia, Laos and Vietnam made up the main cluster, whereas those from Thailand formed a distinct cluster. The China and CMB samples did not cluster specifically with

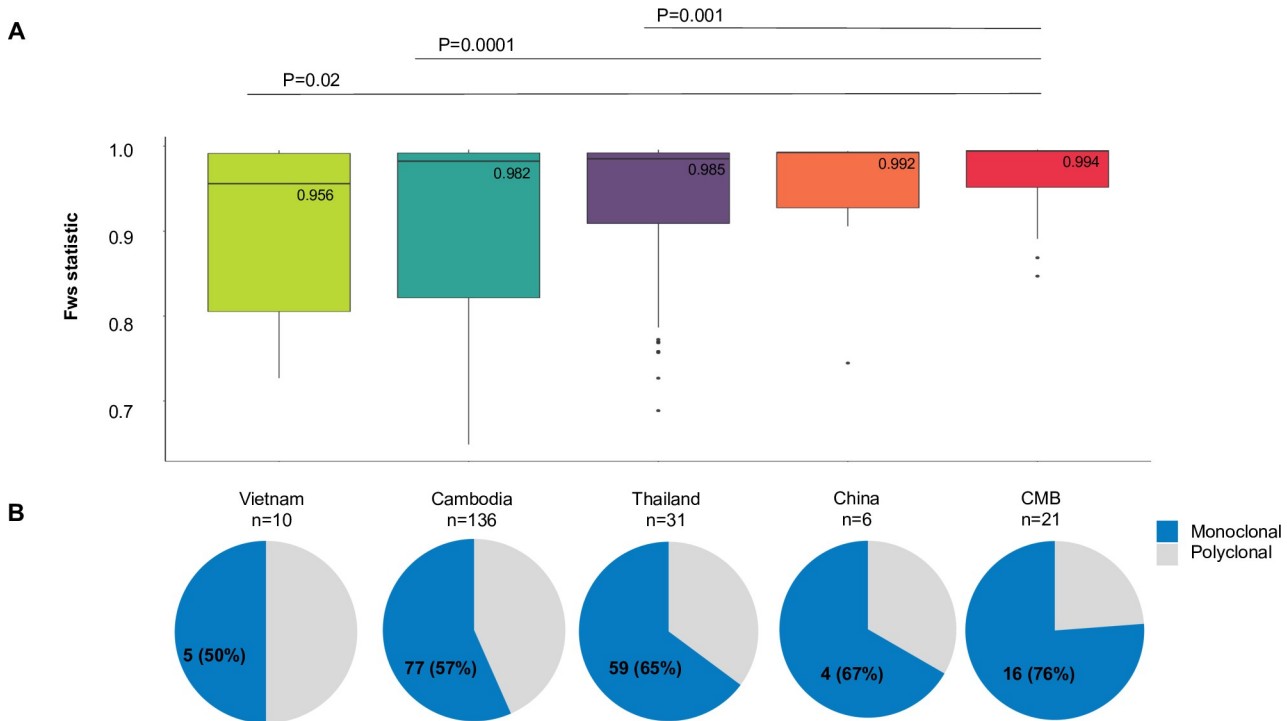

**Fig 2. Clonality within sample sites from the Greater Mekong Subregion.** A) Inbreeding within the GMS countries, as denoted by the $F_{ws}$ statistic. Medians are labelled and *P values* from pairwise Wilcoxon rank-sum test are shown. **B**) Pie charts representing the proportion of monoclonal (blue, $F_{ws} \geq 0.95$) or polyclonal (gray, $F_{ws} < 0.95$) samples for each location ordered as in A.

either of the two major clusters (Fig 3A). Admixture analysis on monoclonal isolates showed that at k = 2 or 3, the 5- and 10-fold cross validation had the lowest errors (S4 & S5 Figs). At k = 2, the CMB and China samples were clearly distinct from the rest of GMS populations (Fig 3B), while at k = 3, this separation still held true, but the major population was divided into two, one being predominantly Thai samples and the other being samples from Cambodia, Laos and Vietnam.

We next evaluated the relationships among the parasite populations in the GMS using phylogenetic analysis. The maximum-likelihood tree provided further support of the results obtained from PCA and admixture analysis (Fig 3C, S6 Fig). Samples from Vietnam and Laos fell into a clade with samples from Cambodia, while most samples from Thailand formed a separate clade. Samples from the Yunnan Province and from the CMB formed a distinct population, albeit it was more closely related to the Thai population. A consensus tree with bootstrap replicates consisting of highly supported nodes showed essentially the same geographical clustering of parasite populations from the GMS (S6B Fig). In addition, repeated analysis with

**Table 1. Pairwise *F*st statistic between countries within the Greater Mekong Subregion.**

|  | CMB | Thailand | Cambodia |
|---|---|---|---|
| Thailand | 0.0891 (0.08821, 0.0900) |  |  |
| Cambodia | 0.0947 (0.0937, 0.0957) | 0.0168 (0.0165, 0.0170) |  |
| Vietnam | 0.1044 (0.1033, 0.1055) | 0.0114 (0.0109, 0.0120) | 0.0020 (0.0016, 0.0024) |

The 2.5% and 97.5% quantiles from 10,000 bootstraps are shown in parentheses.

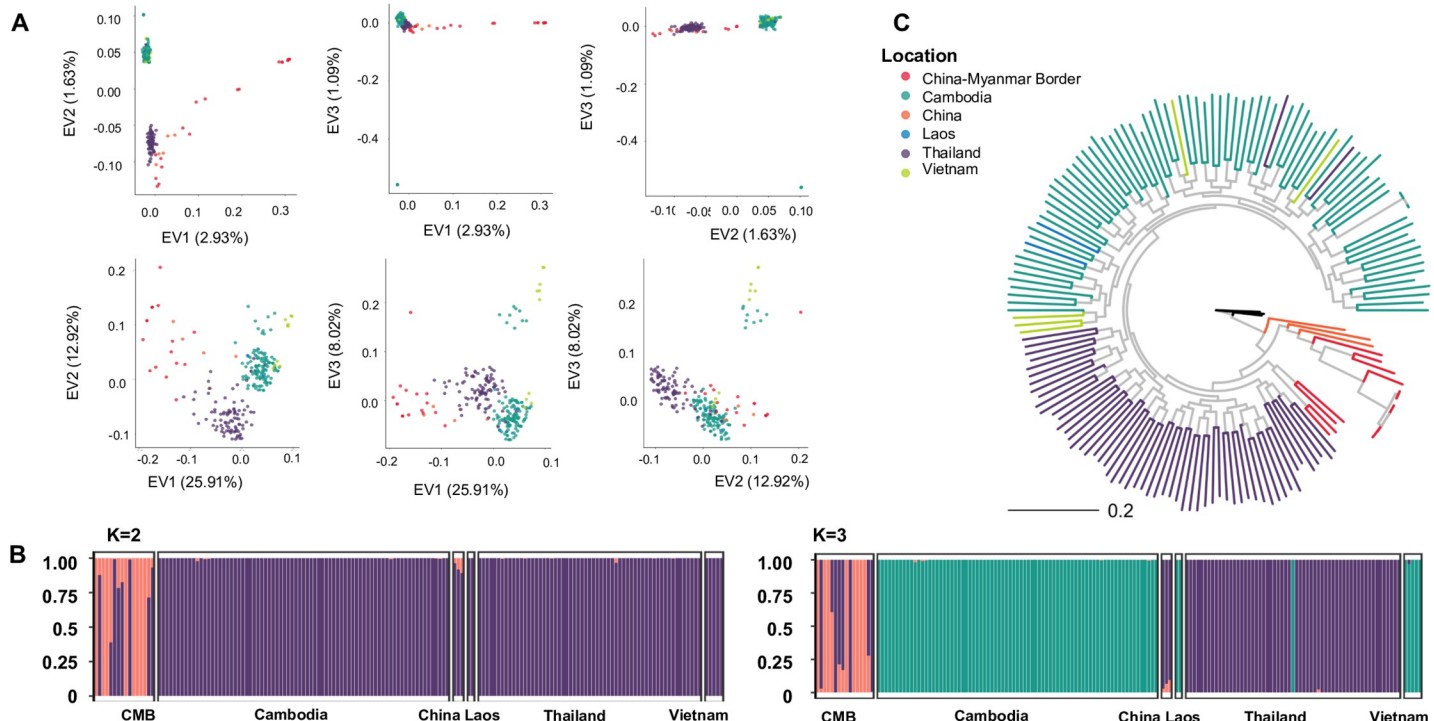

**Fig 3. Population structure of *P. vivax* within the GMS. A)** PCA using SNPs from the entire genome (top panel) or from 36 SNPs with the top $F_{st}$ values which are not in LD (bottom panel) **B)** Admixture analysis using models where k = 2 (left) or k = 3 (right) and each predicted population is denoted by a separate coloration. **C)** Maximum-likelihood tree of monoclonal samples from across the GMS predicted in RAxML. Black lines denote an outgroup from Madagascar.

randomly chosen sets of 5,000 SNPs with no filter on MAF produced trees with reasonably similar clade formation among the GMS parasite populations (S7 Fig).

## Identification of a SNP barcode for separation of parasite populations from the GMS

With the illustration that genome-wide SNPs enable the separation of parasite populations within the GMS, we were interested in developing a SNP barcode that is suitable for practical identification of origins of parasites from the GMS. We performed whole genome scans of the pairwise $F_{ST}$ values in attempt to identify regions which were differentiated between two populations (Fig 4). To determine if a small number of highly differentiated SNPs could reproduce population structure, we extracted the top 10 most highly differentiated SNPs from each pairwise comparison, resulting in 60 total SNPs. Seven of these SNPs were redundant and 17 SNPs were not independent, resulting in 36 SNPs left. We were able to reconstruct a similar population structure to that seen in whole-genome analysis with the remaining 36 SNPs (Fig 3A). In order to see if other published barcodes may be effective on this dataset we tested how many of their suggested SNPs occurred as variable within our dataset. Only 9 SNPs from a 42 global barcode were called within our dataset [30]. For the more specific global barcode recently published by Benavente et al. which included 72 SNPs, we had callable variants at 47 of the positions, and 41 of them were independent [31]. Finally, we tested the 28 and 51 SNP barcodes from Trimarsanto et al., which yielded 19 and 34 independent SNPs, respectively [32]. The SNPs not callable within our GMS dataset were mostly due to their invariability or with a MAF of less than 0.01.

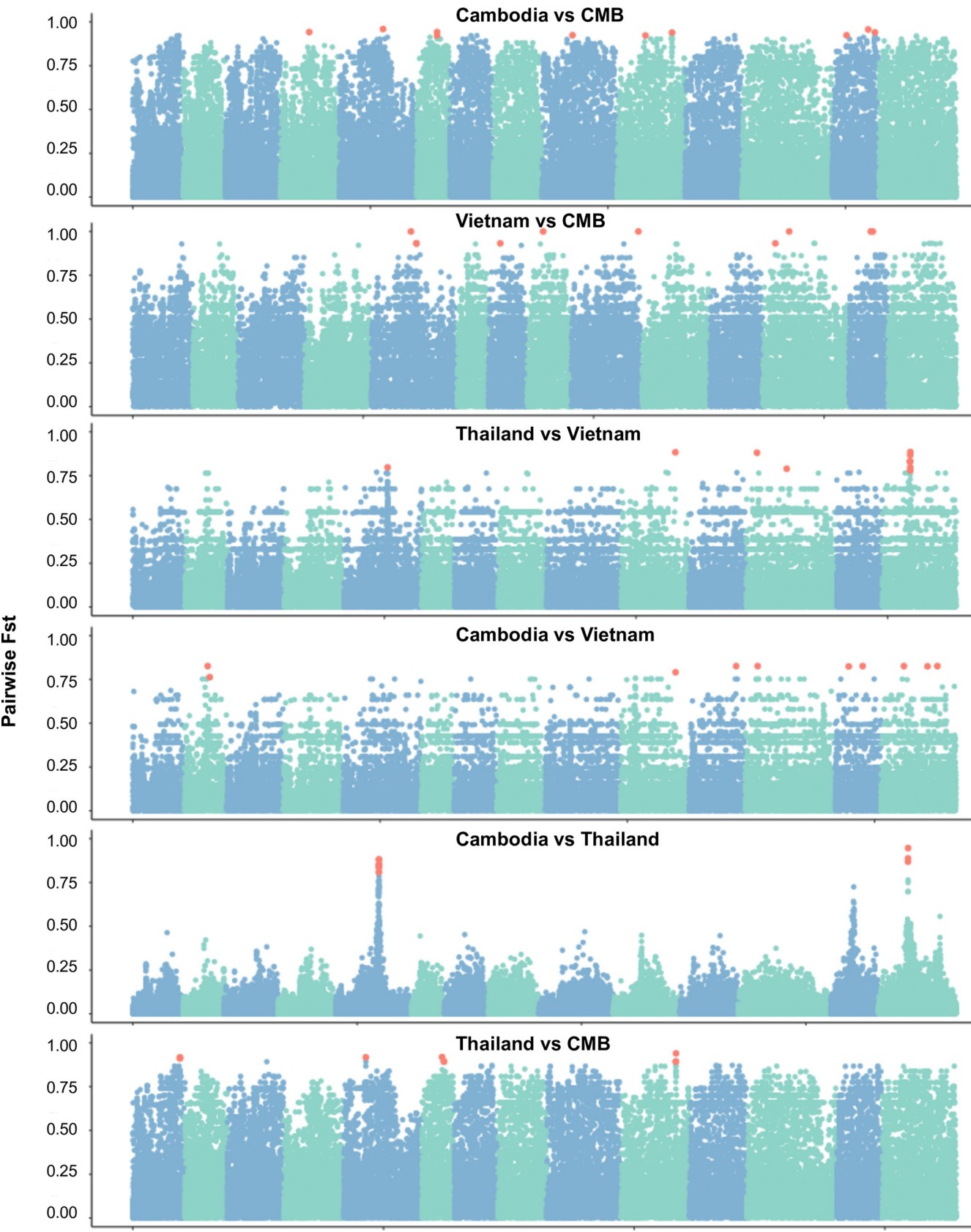

**Fig 4. $F_{ST}$ values across the genome for each pairwise comparison.** Each blue or green section is a subsequent chromosome, with each point being a separate core SNP. Salmon-colored SNPs represent the top 10 SNPs from each pairwise comparison.

## Potential migration patterns of the parasite populations within the GMS

In order to establish the likely patterns of migration and identify potential significant gene flow barriers, we used the program EEMS to estimate migration surfaces. Using the 163 monoclonal samples from the GMS, we identified especially low migration rates around the CMB and within western Cambodia, both supported by posterior probabilities above 0.9 (Fig 5A, S8 Fig). However, regions between Thailand and China did not have reduced effective migration relative to physical distance, nor did areas between Cambodia, Laos and Vietnam.

To determine if genetic separation of the CMB is due to geographic separation, we also established pairwise IBD among each parasite pair to infer the transmission of 263 parasite strains through the GMS. If two parasites shared more than 1% of their genomes as IBD segments uninterrupted by mutation for at least 25 kb, they were considered distantly related, while if two parasites shared more than 90% of their genomes as IBD segments they were considered closely related (Fig 5B & 5C). The IBD network showed two large clusters of distantly related parasites from mostly Thailand and Cambodia, respectively (Fig 5B). The CMB samples formed one large, distinct and tightly connected cluster of shared identity, and a second smaller component of just three samples. The CMB samples were distantly related to other GMS populations, except one sample from Tengchong in western Yunnan Province, also bordering Myanmar. We tested if there was a strong relationship between geographic distance and IBD, noting the highest IBD was found in parasites from the same locations, which very

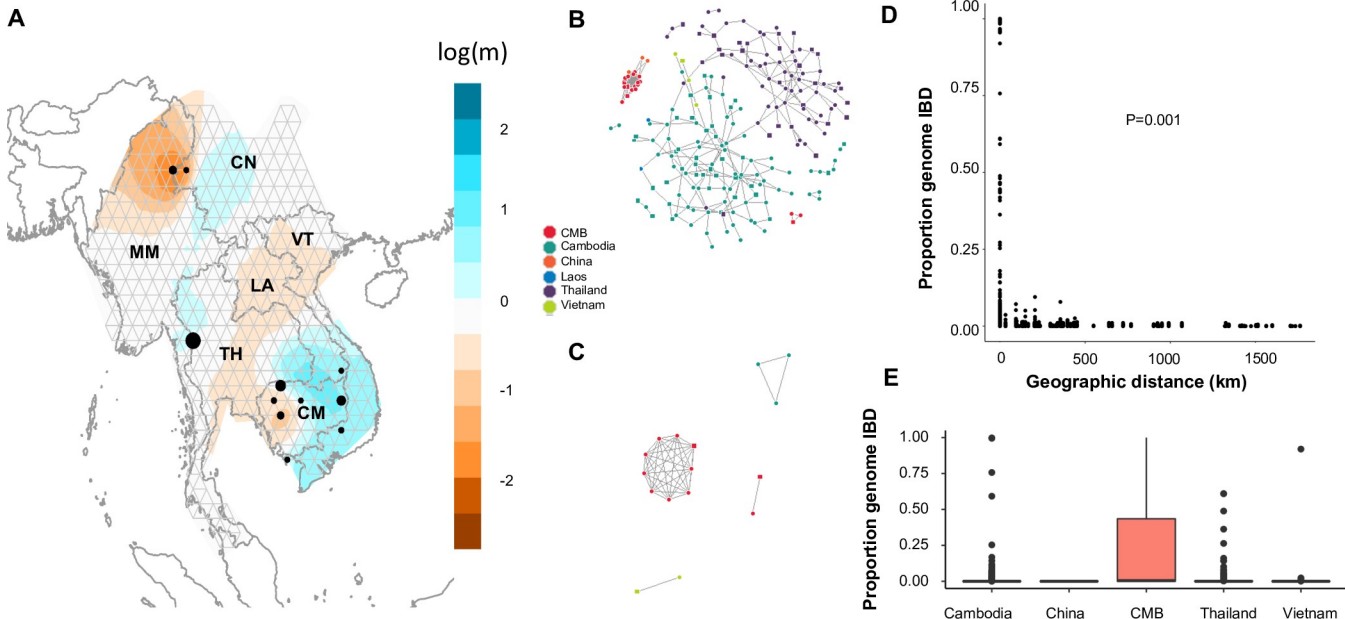

**Fig 5. Relatedness and transmission of *P. vivax* within the Greater Mekong Subregion. A)** Effective migration surfaces wherein each black dot represents one population. Areas which are more orange represent reduced effective migration whereas blue areas represent more effective migration. **B)** IBD network showing connection between samples sharing at least 1% of their chromosomal genome in IBD segments of 25 kb or longer. Squares represent polyclonal samples, while circles represent monoclonal samples. **C)** IBD network showing connection between samples sharing at least 90% of their chromosomal genome in IBD segments of 25 kb or longer. Squares represent polyclonal samples while circles represent monoclonal samples. **D)** Comparison of pairwise shared IBD and distance in km. *P* value is from Mantel testing distance vs IBD. **E)** Pairwise IBD sharing in sample sites within each location. For Thailand and Cambodia, which have multiple sample sites, only samples collected from the same sample sites (distance in km = 0) were included.

quickly decayed as parasites become further apart (Fig 5D). Mantel testing suggested a correlation between distance and IBD ($P < 0.001$), which was confirmed by Spearman correlation between IBD and distance in kilometers ($P < 2.2e\text{-}16$). However, there were also large variations of pairwise IBD values even at a distance of 0 km, suggesting the existence of genetically distinct parasites within a single location.

Among parasites sharing high identity (>90%), two clusters were identified within the CMB, one consisting of 9 parasites and the other 2 parasites (Fig 5C). In contrast, among the rest of the GMS only two connected components were identified sharing 90% identity, one consisting of 3 samples from Cambodia and the other consisting of 2 samples from Vietnam. Because of the disproportionate number of highly-related parasites within the CMB population and the spread of IBD values in 0 km pairs, we gathered pairwise IBD among the same-site parasites from each population with over 5 isolates of high-quality WGS (Cambodia, China, CMB, Vietnam and Thailand) (Fig 5E). The average pairwise IBD within the same-site samples in Cambodia, Thailand, Vietnam and the CMB were 0.004, 0.002, 0.021 and 0.229, respectively, with maximum pairwise IBD values of 0.997, 0.610, 0.919 and 1. The Tengchong (China) samples had no detectable recent IBD. We replicated analyses comparing distance and IBD values using the program hmmIBD with two populations and found comparable distributions (S9 Fig).

To test the influence of potential clonal expansion on population structure we chose a representative sample from each of the 4 clusters with IBD sharing above 0.9 (S2 Table). Ultimately, 9 samples from the China-Myanmar border, 2 from Cambodia and 1 from Vietnam were removed. After this correction, countries were still separable in PCA using both the full genome SNPs and the 36 SNP barcode and the $F_{ST}$ averages between populations were still higher between the CMB and other GMS populations (S10A Fig, S4 Table). However, the degree of differentiation between populations became smaller with no pairwise $F_{ST}$ averages surpassing 0.05, and the lowest error admixture modeling consisting of just one population (S10B Fig). Admixture analyses at two populations revealed the CMB in a population with China and Thailand, while at K = 3 CMB parasites largely were in its own population though with some sharing with Cambodia and China (S10C Fig).

## Genetic differentiation of antifolate resistance genes

To explore how the CMB population differs from elsewhere in the GMS, we scanned high-$F_{ST}$ SNPs for stretches where the populations displayed high degrees of differentiation. One stretch between 1,070,634 and 1,087,230 on chromosome 5 stood out, wherein 17.0% (9/53) of the high-$F_{ST}$ SNPs were located (Fig 4). The five genes annotated in this region include the mRNA-binding protein *Puf2*, the *bifunctional dihydrofolate reductase-thymidylate synthase* (*dhfr-ts*), the *LETM-1like protein* and 2 conserved proteins with unknown functions. A second region on chromosome 14 between base pairs 1,231,865 and 1,270,401 contained 6 (11.3%) high-$F_{ST}$ SNPs. This section contains 8 genes including the genes for *CARM1*, *CCR4-associated factor 1*, *cullin-1*, a mitochondrial carrier protein, two conserved *Plasmodium* proteins, and *hydroxymethyldihydropterin pyrophosphokinase-dihydropteroate synthase* (*dhps*).

Further analysis of *pvdhfr* gene in samples from the CMB, Thailand and Cambodia identified 8 non-synonymous mutations after removing mutations unique to a single sample (S1 Data). All but four mutations in the CMB, Thailand and Cambodia populations were fixed or close to fixation with a MAF of 0, 0.01 and 0.007, respectively. The four mutations which were not fixed are F57I/L, S58R and T61M. The mutations for F57I/L were completely absent in Cambodia, but were present in 85.7% of Thailand parasites, and 33.3% of the CMB parasites, respectively. T61M was absent in Cambodia, but was present in 85.6% of the Thailand

population, and 33.3% of the CMB population. Finally, S58R was fixed in Thailand and Cambodia, but present in only 38.1% of the CMB samples. Seven non-synonymous mutations were identified in the *pvdhps* gene, 3 with a MAF above 0.05 in at least one population (S1 Data). For the A553G substitution, Thailand samples were nearly fixed for the mutant allele, whereas Cambodia samples were all wild-type allele, and the CMB samples had 28.6% A553G. For K512M/E, Cambodia samples were all wild-type alleles, while nearly 5% of the CMB samples had the 512E mutation. In Thailand, 4% and 32.6% samples had K512M and S382A, respectively. At G383A, Thailand samples were almost all wild type, while the CMB and Cambodia had a mixture of the wild-type and mutant alleles.

Because chloroquine is the first-line of treatment for *P. vivax*, we also explored non-synonymous mutations in the *pvmdr1* gene. Seven non-synonymous mutations were identified in *pvmdr1*, all of which had a MAF over 0.05 (S1 Data). Among those were Y976F and F1076L, which previously were associated with chloroquine resistance [40]. All CMB samples had the wild-type Y976 allele, while Cambodia and Thailand had 64% and 16.4% of the mutant 976F allele, respectively. The mutant 1076L allele was present in 51.6, 67.0 and 86.1% of the Thailand, CMB and Cambodia samples, respectively.

## Selective sweeps within the GMS

We scanned the parasite genomes from the CMB for chromosomal regions with suspected selective sweeps. Out of concern for stretches of homozygosity induced in filtering, we added additional regional masks before undergoing scans of selective sweeps which eliminated 28 genes on chromosomes 1–14 for which at least one base pair was masked for high coverage variability (S3 Table). From 30,689 total SNPs analyzed, 307 sites were identified to be potentially involved in a selective sweep, each having at least two SNPs within the top 1% of absolute nSL values within a 10 kb region, which were classified as extremely high nSL values (Fig 6A, S2 Data). This led to the identification of 157 genes located near the high nSL SNPs, including genes encoding phosphatidylinositol-4-phosphatase 5-kinase (PIP5K), cytoadherence-linked asexual protein 7 (CLAG7), an ABC-1 family kinase (ABCk2), multidrug resistance-associated protein 2 (MRP2) and vacuolar type H+ pumping pyrophosphatase (VP1). To see which genes were selected differentially between the CMB and the rest of the GMS, XP-EHH analysis was performed, leading to the identification of 995 top or bottom 0.5% SNPs in terms of normalized XP-EHH score and linked within 10 kb of another high or low XP-EHH value (Fig 6B, S2 Data). These are mapped near 297 genes including *eukaryotic translation initiation factor 2-alpha kinase 1* (*eIF2-γ*), *MRP2*, and *ABCk2*. Stretches of extremely low XP-EHH regions occurred around *multidrug resistance-associated protein 1* (*MRP1*), *AP2 domain transcription factor AP2-I*, *ABC transporter I family member 1* (*ABCI3*), *apical membrane antigen 1* (*AMA1*), *N-acetylglucosaminyl-phosphatidylinositol de-N-acetylase* (*PIGL*), and *VP2*. In addition, within- and between-population comparisons identified 41 genes near SNPs with both extremely high XP-EHH and extreme nSL, including *MRP2*, *gametocyte development protein 1* (*GDP1*), *AMA1*, *PIGL*, *ABCk2* and *VP2*. Province-level XP-EHH comparisons between the CMB samples and those from Tak Province, Thailand, and from Oddar Meanchey, Cambodia were consistent with those from the entire GMS with 86 of 127 genes identified in at least one of the province-level comparisons as well as the whole-GMS comparison (S2 Data).

To illustrate conservation among genes potentially under selection within and between populations, we performed IBD analysis at the genomic regions encoding MRP2, CLAG7 and VP2, which are situated near high nSL signals (Fig 6). In each case, the largest connected component consisted of the CMB, consistent with high global IBD (Fig 5C, 5D and 5E). Ten components of IBD-sharing samples were identified at the *CLAG7* locus, but none between

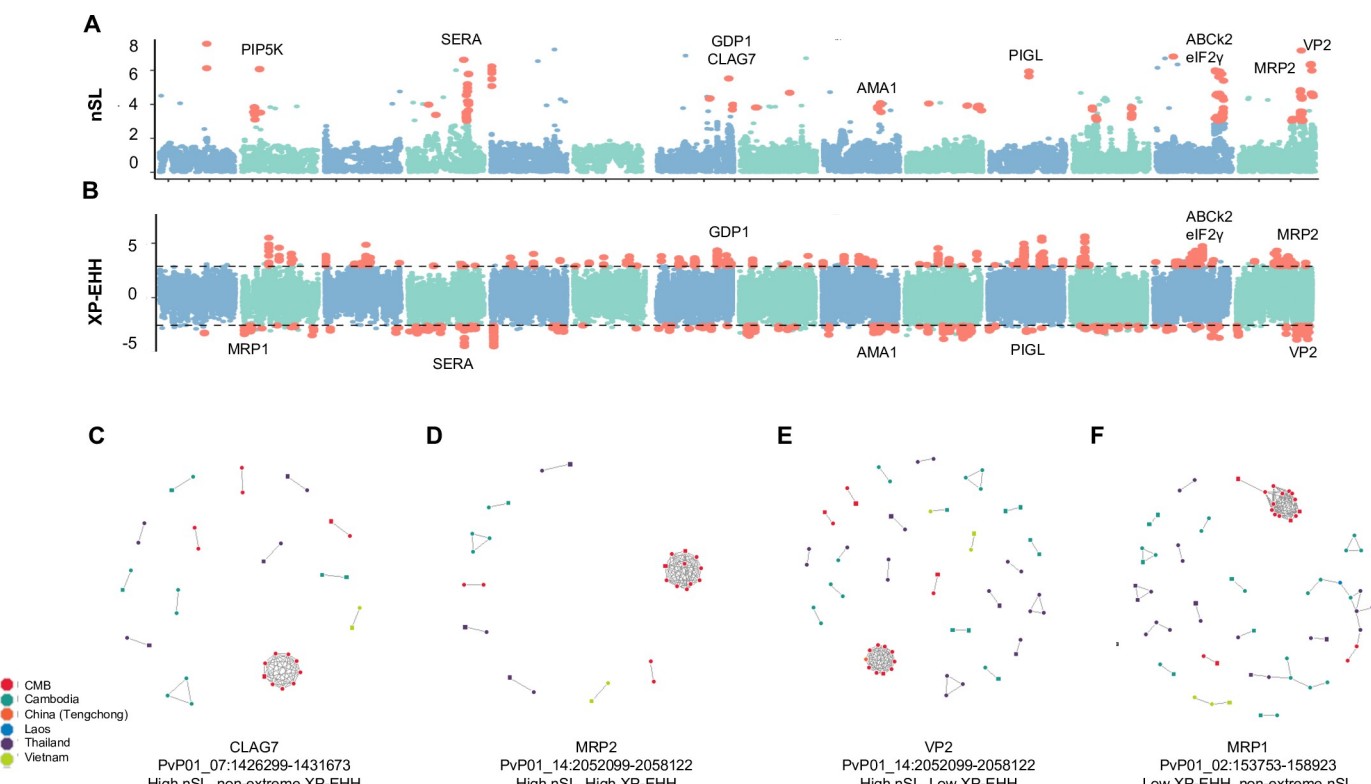

**Fig 6. Signals of selective sweeps within the China-Myanmar border parasite population. A)** Signatures of selection across the genome wherein alternating blue and green colors denote separate chromosomes and red dots represent points which are in the top 1% and also have at least one neighboring SNP within 10 kb which is also in the top 1% of values. **B)** XP-EHH statistic. **C-F)** IBD networks for CLAG7 (**C**), MRP2 (**D**), VP2 (**E**), and MRP1 (**F**). Genes were chosen to represent different combinations of XP-EHH and nSL result status. Dots of different colors represent parasites from different countries (regions), respectively. Polyclonal samples have a square shape, while monoclonal samples have a round shape.

samples from two countries (Fig 6C). At the *MRP2* locus, IBD sharing between non-CMB GMS samples was low with only four groups of nine IBD-connected samples emerging (Fig 6E). Meanwhile, CMB samples were highly connected at the *MRP2* locus with 15 samples forming 3 connected components. This result and the nearby high XP-EHH signals are consistent with more conservation of *MRP2* within the CMB than elsewhere (Fig 6B). At *VP2*, CMB samples formed 4 components of 15 samples, similarly to *CLAG7* and *MRP2* (Fig 6F). However, other GMS samples were more connected in *VP2* than in other genes and formed 16 clusters of IBD sharing, consistent with a lower XP-EHH value (Fig 6B). Three of the non-CMB clusters contained more than two samples, and two clusters displayed IBD sharing between samples from two countries suggesting more cross-GMS conservation of *VP2* than other genes tested. Because the high degree of global IBD sharing within the CMB caused all genes to have strong IBD compared to other regions, we also performed IBD analysis on the *MRP1* gene, which had a low XP-EHH value but did not have a high nSL value (Fig 6F). Even in this case, the CMB had a high degree of IBD-sharing with 14 samples forming one large connected component, 2 others forming a second, and 2 more forming a cross-population component with samples from Thailand, Cambodia and Laos.

For genes with functional implications in drug resistance, we extracted non-synonymous allele frequency data for cross-population comparisons (S1 Data). There were 23 non-synonymous mutations within *MRP2*, of which 19 were fixed in the 21 CMB samples (S1 Data). In Thailand, 4 alleles were fixed with all others having a MAF of 0.01 or higher. In Cambodia, 5

alleles were fixed and one other had a MAF below 0.01. In the case of *VP2*, two non-synonymous mutations L451F and A609V were absent in the CMB samples, while Cambodia and Thailand had 0.01 and 0.007 frequencies of non-reference alleles at both positions.

## Discussion

Border malaria within the GMS is a focal point for regional malaria elimination efforts. The CMB, specifically, has predominantly *P. vivax* malaria with recent outbreaks detected from malaria surveillance [7]. Here we obtained whole genome sequences from 23 clinical samples from the CMB, of which we used information from 21 high-coverage isolates to characterize *P. vivax* populations in this region. We found that parasites from this region had significantly higher levels of clonality than parasites from other regions of the GMS. While the CMB parasites appeared to be genetically isolated from other populations in the GMS, they also showed a significantly higher degree of IBD, suggesting the result of recent clonal expansion. By scanning chromosomal regions for signatures of selection, we identified genes that are potentially associated with drug resistance.

Multiple clonal infections are common in *P. vivax* even in low-endemicity areas, which may result from new and recurrent infections. To a certain degree, multiplicity also reflects the levels of endemicity, and is biologically important in terms of genetic recombination in mosquitoes. Additionally, multiplicity of infection is a notable concern for WGS studies because it can result in genotyping errors [41]. The CMB had a higher frequency of monoclonal infections than other regions of the GMS (Fig 2). Traditionally, an $F_{ws}$ value above 0.95 is suggestive of monoclonal infection, and we found that over ¾ of samples from the CMB surpassed this threshold (Fig 2B). Additionally, the overall distribution of $F_{ws}$ values within the CMB was significantly higher than those in Cambodia or Thailand (Fig 2B). $F_{ws}$ values within more closely related parasites are typically higher, pointing to the possibility of either lower local genetic diversity or lower transmission rates [16,41]. The high degree of pairwise IBD within the CMB parasites further suggests transmission of a limited number of strains (Fig 5E), which was consistent with the vivax outbreak occurring during the period of sample collection [7]. This is reminiscent of the rapid clonal expansion found in the malaria pre-elimination Malaysia [42]. Our data support that prior to the outbreak there was relatively low transmission in the CMB with a limited number of predominant circulating strains.

Except for parasites from the CMB, *P. vivax* populations within the GMS appeared to be more or less panmictic, which may reflect the absence of significant gene flow barriers. Parasite populations from Thailand, Cambodia, Laos and Vietnam showed little population differentiation, whereas they were more divergent from the CMB parasites (Table 1, S4 Table). Phylogenetic trees placed the samples from CMB and Tengchong, Yunnan in a separate clade, which was more closely related to parasites from Thailand than parasites from elsewhere (Fig 3C, S5 Fig, S6 Fig). Such a relationship was further supported by admixture models, though the relationship between Tengchong and the CMB samples was less apparent than between Tengchong and Thailand samples (Fig 3B, S4 Fig, S5 Fig). This discrepancy could be an artefact of different methodologies, since higher k-value admixture analyses demonstrated more clear population sharing between Tengchong and the CMB, while bootstrap values for the clade between the CMB and Tengchong were low and not as apparent in a consensus tree made of only high-confidence clades (S6 Fig, S5 Fig). Our results agree with earlier findings that *P. vivax* samples from Tengchong were divergent from Cambodian parasites [17,20]. Further, PCA results also consistently showed separation between the parasite samples from Thailand and Cambodia, while parasites from Laos and Vietnam repeatedly situated within the Cambodia grouping (Fig 3A, Fig 5). The results from analysis of genome-wide SNPs were

encouraging, given the significance of differentiating parasite isolates for tracking parasite origins in the final phase of malaria elimination.

A limited number of microsatellite markers were previously able to distinguish *P. vivax* parasites from western and eastern Thailand [8]. However, while many global barcodes exist, a robust barcode of a small number of SNPs tailored to the GMS is still lacking. A 42-SNP barcode exists for global population differentiation [30]. In this region, however, we found a low number of SNPs that were callable, suggesting that there is no variation within these sites compared to the PvP01, or at least no variation which exceeds a MAF of 0.01. Previously, a 72-SNP barcode published by Benavente et al. was shown to perform better than the 42-SNP barcode for country-level differentiation, though it struggled to separate out closely related parasite populations within the GMS with high resolution and appears to have largely fixed alleles within this region for many of the loci [31]. Our inability to call more than 41 independent variants from the 72-SNP barcode supports this finding. Finally, the recently introduced 28- and 51-SNP barcodes reported the lowest performance between the GMS countries Vietnam and Cambodia [32]. Additionally, when we attempted to implement the 28- and 51-SNP barcodes, only 19 and 34 SNPs, respectively, were callable SNPs within this dataset, suggesting many of the SNPs are not sufficiently polymorphic within the GMS. In an effort to identify a set of regionally-specific SNPs, we screened SNPs with the highest pairwise $F_{ST}$ between populations and found that 36 SNPs allowed reproduction of the parasite population structure built from genome-wide SNPs. Importantly, the barcode succeeded at separating most countries, including Vietnam, especially when polyclonal samples were removed (S3 Fig). One limitation to our barcode is the lack of samples from inland Myanmar as well as few samples from Laos which may result in a lack of effectiveness in these parts of the GMS. Accordingly, the Laos samples were included in a cluster with Cambodia, rather than separated out (Fig 3A). Additionally, our cross comparison between barcodes was limited to variability within our dataset. The usefulness of this new *P. vivax* barcode for distinguishing parasite populations in the GMS requires future evaluation.

Although the separation of CMB parasites from the rest of the GMS is corroborated by a significant correlation between geographic distance and IBD (Fig 5), the correlation was not as strong as has been demonstrated for *P. falciparum* from the same region [20]. Previously, *P. vivax* was identified as having higher diversity with a higher effective population than *P. falciparum*, possibly due to less selective pressure and the ability of parasite to relapse to reintroduce genetic traits resulting in fewer genetic bottlenecks [14,43]. A larger effective population size with fewer localized bottlenecks could explain the reduced tendency towards isolation by distance in *P. vivax* witnessed here. The only areas with decreased migration rates that were of high confidence were the CMB and the central-eastern part of Cambodia, results that mimic findings from *P. falciparum* [20]. Yet, the conclusion that the CMB poses a barrier to malaria transmission appears inconsistent with microsatellite data which suggests directional cross-border migration from Myanmar to China [9,10]. This discrepancy may be due to unidirectional migration as EEMS does not consider migration directionality. Notably, the separation between the CMB and China became less pronounced upon the removal of the high-IBD cluster (S10 Fig), though population sample sizes after this correction are too low to give definitive conclusions. Regardless, our analysis suggests that parasite migration may be an important factor for genetic mixing of parasites across considerably large distance in the GMS, emphasizing the need for enhanced monitoring of parasite introduction by migrating human populations. Since removal of the high-IBD cluster from the CMB samples reduced differentiation from Cambodia, Thailand and Vietnam, parasite introduction from the southern or eastern parts of the GMS seems unlikely (S10 Fig, S4 Table). Further, in the study area at the CMB, which is difficult to access, earlier studies indicated local transmission of *P. vivax* as compared to

introduced *P. falciparum* cases associated with recent travel histories [44]. Thus, the origins of the CMB parasites potentially under clonal expansion may be local or from nearby endemic areas introduced with the migrant populations who came to the CMB areas while escaping the Kachin civil wars [45]. It is also noteworthy that no whole-genome *P. vivax* samples are currently available for inland Myanmar which reduces our ability to track parasites that may have come from other parts of the country. Drug pressure frequently selects for certain parasite strains [46,47], and could influence clonal expansion of these strains.

Certain regions of the genome have higher degrees of differentiation within and among the GMS populations (Fig 4), suggestive of local selection. Particularly, strong signals of selection were present in *pvdhfr* and *pvdhps* genes associated within resistance to pyrimethamine and sulfadoxine, respectively. Previous studies have identified multiple alleles surrounding these genes which vary geographically [13,14,48,49]. Despite this, previous studies in the GMS have largely supported fixation of the 58R/117N mutation to which antifolate resistance is attributed [48–50]. The antifolate drugs have been used as replacement drugs of chloroquine for treating *P. falciparum* in some areas of the GMS, which may have posed collateral selection in *P. vivax* since co-infections by *P. vivax* and *P. falciparum* often occur [7,13]. In addition, the antifolate drugs have been used in Southeast Asia for malaria prophylaxis [51]. Differing regional drug histories may be responsible for the different prevalence of resistance-conferring mutations in different *P. vivax* populations. Our results support previous reports of fixation 58R/117N within Cambodia and Thailand, as well as the higher frequency of quadruple mutants towards the Thailand-Myanmar border and double mutants closer to the Cambodia-Thailand border [48], however 58R is present in less than half of the CMB samples (S1 Data). We also examined the *pvmdr1* gene and found that the CMB population was completely wild-type at the Y976 locus, but over half had the 1076L mutation. However the contribution of the *pvmdr1* mutations to chloroquine resistance in *P. vivax* remains inclusive [52]. Nonetheless, geographical variation in allele frequencies of these mutations with potential impacts on drug resistance between countries of the GMS suggests that they might be under drug selection, as documented in *P. falciparum*.

In addition to *pvdhfr*, *pvdhps* and *pvmdr1*, we explored parasite genomes for signs of purifying selection as well as regions under different selective pressure between the CMB population and parasites from elsewhere in the GMS using complementary programs. Previous comparisons between scans for selective sweeps determined that iHS (closely related to nSL) is effective at determining soft sweeps in response to selective pressures [36]. Importantly, the high degree of IBD between samples means that even genomic regions, which were not located to SNPs within the top 1% of the nSL values (e.g., around *mrp1*), may have a high degree of IBD within the CMB samples (Fig 6C). Some genes and regions (e.g., *mrp1*, *mrp2*, *vp2*), identified by both nSL and XP-EHH, may be related to drug resistance. *Pvmrp1* appears to be under less selective pressure within the CMB than elsewhere in the GMS based on the XP-EHH values (Fig 6B & 6E). It is noteworthy that the *P. falciparum mrp2* ortholog is associated with resistance to mefloquine and chloroquine [53–55]. *Pvmrp2* was found to be especially diverse, but the non-synonymous alleles were nearly entirely fixed in the CMB population (S1 Data). For the vacuolar pump *pvvp2* gene, a high nSL value accompanied by a negative XP-EHH value signals purifying selection within the GMS. *Pfvp2* has previously been shown in *P. falciparum* to be associated with chloroquine resistance, though this might be due to genetic linkage to *pfcrt* [56]. Therefore, mutations in these genes could be resulted from selection by chloroquine, given chloroquine/primaquine has remained the frontline treatment for vivax malaria in the GMS. It would be interesting to see whether the mutations in these genes are correlated with drug

resistance, as there is clear evidence of emerging chloroquine resistance in the northeastern Myanmar *P. vivax* population [46,47].

The GMS is pursuing malaria elimination and malaria in most areas has experienced dramatic declines in recent years. Malaria in many parts has, therefore, been limited to some international border regions and separated by malaria-free areas. Monitoring the parasite population evolution within these pockets as well as introduction becomes an important task for the final phase of elimination. Here we used genomic data acquired from clinical parasite isolates in the CMB and other regions of the GMS to compare the genetic diversity and population differentiation of the *P. vivax* parasite in the GMS. Our findings indicate that parasite populations from the eastern GMS were largely undifferentiated, whereas parasites from the CMB showed substantial divergence from the rest of the GMS populations. There was also evidence showing lower levels of recombination and clonal expansion of CMB parasites, which are co-incidental with *P. vivax* outbreaks and emerging resistance to chloroquine. The genetic background of *P. vivax* in this region demonstrated that genes linked to antifolate resistance were not fixed as they were in other parts of the GMS, suggesting drug resistance patterns and thus optimal treatment may differ compared to other GMS regions. In addition to mutations in antifolate resistance genes, we identified signatures of selection on genes potentially associated with drug resistance which may guide future efforts in understanding this population. Finally, we specially curated a set of SNPs for population differentiation in the GMS which may be useful for monitoring progress of malaria control in this region.

## Supporting information

**S1 Table. Alignment statistics for 23 CMB *Plasmodium vivax* field samples.**
(PDF)

**S2 Table. Sample statistics for each sample used in study.**
(XLSX)

**S3 Table. Regions in the genome removed due to high chances of mismapping.** Italicized regions on the right were only removed during selection scans, where excessive loss of polymorphism from removal of high-CV SNPs may result in selection bias.
(PDF)

**S4 Table. Fst values corrected for structure by condensing of IBD clusters.**
(PDF)

**S1 Fig. Coefficient of variation (CV) among sliding windows on each chromosome.** CV = 1 is marked in red to show cutoff for pruning due to high likelihood of mismapping. For chromosomes where CV in some case rose above 5, values above 5 were excluded from plotting, but were still masked.
(TIF)

**S2 Fig. Prediction of functions of high-quality SNPs from SNPeff.** Potential effects from each variant are categorized into various functional (top panel) and regional (bottom panel) categories. Variants may have more than one effect.
(TIF)

**S3 Fig. Principal component analysis performed with monoclonal samples. A**) All SNPs, **B)** High-$F_{ST}$ barcode only.
(TIFF)

**S4 Fig. Admixture model quality. A)** K-value cross validation scores for admixture analysis at 5 and 10 folds. **B&C)** Standard error across 1000 bootstraps for **(B) k = 2** and **(C)** k = 3 admixture models.
(TIF)

**S5 Fig. Admixture analyses for 1–10 populations within the GMS.** K value gets ascendingly larger going from top to bottom on the left than the right column.
(TIF)

**S6 Fig. RAxML Maximum-Likelihood trees for Greater Mekong Subregion with bootstrap confidence. A)** RAxML maximum likelihood tree where nodes are pictured with pie charts representing bootstrap values (wherein darker circles hold higher confidence) from 1000 bootstraps. Tree is based on 40,008 high-quality internal SNPs. **B**) Consensus tree based on 1000 bootstraps.
(TIF)

**S7 Fig. RAxML Maximum-Likelihood trees for the Greater Mekong Subregion with randomly selected sets of 5000 SNPs.** Each tree was constructed with a new set of 5000 random SNPs regardless of MAF. Colors: Salmon (CMB), Orange (China), Lavendar (Thailand), Teal (Cambodia), Blue (Laos), Green (Vietnam). Nodes are pictured with pie charts representing bootstrap values (wherein darker circles hold higher confidence) from 1000 bootstraps. Tree is based on 40,008 high-quality internal SNPs.
(TIF)

**S8 Fig. EEMS migration data with posterior probability $> 0.9$.**
(TIF)

**S9 Fig. Site-based IBD analysis using hmmIBD.** Hmm-IBD estimates were made under the assumption of two populations which we defined by those in the CMB and those not in the CMB as supported by admixture analysis. **(A)** Geographical distance in kilometers compared to proportion of genome predicted to be IBD**. (B)** IBD-sharing among parasites from the same sample sites (distance = 0) in each location.
(TIF)

**S10 Fig. Population structure analyses with IBD clusters condensed. A)** Principle component analysis with full SNP set (top) and barcode only (bottom). **B)** Admixture cross-validation errors at various k-values. **C)** Admixture analyses at k = 2 and k = 3.
(TIF)

**S11 Fig.  XP-EHH for the China-Myanmar border using A)** the rest of the GMS as reference, **B)** Oddar Meanchey, Cambodia as reference, and **C)** Tak Province, Thailand as reference. Red points represent pairs of SNPs within either the top or bottom 0.5% of all SNPs for XP-EHH value.
(TIF)

**S1 Dataset. Allele Frequency data for 53 SNPs with high FST (sheet 1) and selected genes (sheet 2).** SNPs without linkage disequilibrium used in barcode are bolded.
(XLSX)

**S2 Dataset. nSL and iR top hits with predictions.**
(XLSX)

## Acknowledgments

The authors express gratitude to Drs. Margarita M. López-Uribe, Maciej Boni, Santhosh Giririjan, and Runze Li for their thoughtful comments on experimental design, data analysis and

visualization. We are also grateful to the Penn State Genomics Core Facility for their assistance with DNA sequencing and the University of South Florida Research Computing for use of computational resources.

## Author Contributions

**Conceptualization:** Awtum M. Brashear, Liwang Cui.

**Data curation:** Awtum M. Brashear, Yubing Hu, Yuling Li, Yan Zhao, Zenglei Wang.

**Formal analysis:** Awtum M. Brashear.

**Funding acquisition:** Liwang Cui.

**Investigation:** Awtum M. Brashear, Qi Fan, Yubing Hu, Yuling Li, Yan Zhao, Zenglei Wang, Jun Miao.

**Methodology:** Awtum M. Brashear, Yaming Cao, Alyssa Barry, Liwang Cui.

**Project administration:** Liwang Cui.

**Resources:** Qi Fan, Yubing Hu, Yuling Li, Yan Zhao, Yaming Cao, Liwang Cui.

**Supervision:** Jun Miao, Alyssa Barry, Liwang Cui.

**Visualization:** Awtum M. Brashear.

**Writing – original draft:** Awtum M. Brashear, Yaming Cao, Jun Miao, Alyssa Barry, Liwang Cui.

**Writing – review & editing:** Awtum M. Brashear, Alyssa Barry, Liwang Cui.

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
