## [Decision Letter · Decision Letter 0]

29 Apr 2020

Dear Dr. Cui,

Thank you very much for submitting your manuscript "Population genomics identifies a distinct Plasmodium vivax population on the China-Myanmar border of Southeast Asia" for consideration at PLOS Neglected Tropical Diseases. As with all papers reviewed by the journal, your manuscript was reviewed by members of the editorial board and by several independent reviewers. The reviewers appreciated the attention to an important topic. Based on the reviews, we are likely to accept this manuscript for publication, providing that you modify the manuscript according to the review recommendations. 

Sincerely,

Kamala Thriemer

Guest Editor

Hans-Peter Fuehrer

Deputy Editor

Reviewer's Responses to Questions

**Key Review Criteria Required for Acceptance?**

**Methods**

-Are the objectives of the study clearly articulated with a clear testable hypothesis stated?

-Is the study design appropriate to address the stated objectives?

-Is the population clearly described and appropriate for the hypothesis being tested?

-Is the sample size sufficient to ensure adequate power to address the hypothesis being tested?

-Were correct statistical analysis used to support conclusions?

-Are there concerns about ethical or regulatory requirements being met?

Reviewer #1: This manuscript describes 21 new genome-wide sequences of Plasmodium vivax isolates from the China-Myanmar border (CMB) in Asia. Newly generated genome data are compared with other ,publicly available sequences from Cambodia (n = 136_, Thailand (n = 91), Vietnam (n = 10), Yunnan Province of China (n = 5), and Laos (n = 2) to provide a comprehensive overview of the population genomics of this species in the Greater Mekong Region (GMR). Methods used for generating and analyzing genome data are entirely appropriate and up to date.

Reviewer #2: (No Response)

Reviewer #3: L188-189 What is the reason of including polyclonal samples? Whether Fst and PCA show similar results if only monoclonal samples are included.

**Results**

-Does the analysis presented match the analysis plan?

-Are the results clearly and completely presented?

-Are the figures (Tables, Images) of sufficient quality for clarity?

Reviewer #1: Results are clearly presented and show relatively little gene flow between CMB parasites and those from neighboring areas in GMR. Interestingly, instances of clonal propagation of CMB parasites have been observed.

Reviewer #2: (No Response)

Reviewer #3: L288-290 It's not clear why compare Fws values between GMS countries? Fws values are purposed to identify mono vs. polyclonal samples. The comparison of polyclonal rate makes sense among regions, but it's not meaningful to compare Fws.

I suggest insert a Table that show complete information (including position, gene function, mutant frequency in each of the five countries/regions) of the 36 high-performance SNPs. 

Figure 1 – Maps need better resolution and site labels to navigate the readers. Likewise, for other figures that need higher quality

**Conclusions**

-Are the conclusions supported by the data presented?

-Are the limitations of analysis clearly described?

-Do the authors discuss how these data can be helpful to advance our understanding of the topic under study?

-Is public health relevance addressed?

Reviewer #1: Discussion is comprehensive and appropriate. I would like to see a little more about the epidemiological implications of these findings.

Reviewer #2: (No Response)

Reviewer #3: L526-527 Any information on the malaria history that the CMB cases are relapse and a possible correlation between monoclonality and relapse? 

Along with this point, in L561 - any information on possible relapse cases among the samples or references that support the statement that relapses might reduce the correlation between geographic distance and IDB. 

L539-544 While the tree shows monophyly of CMB and Yunnan samples, both K2 and K3 admixture plots indicate distinct clusters for the two regions. Need to provide explanation.

**Editorial and Data Presentation Modifications?**

Reviewer #1: No major revisions are suggested. I just would like to see a more epidemiology-oriented general discussion about the implications of the population genetic findings for malaria control and elimination in the region.

Reviewer #2: (No Response)

Reviewer #3: L143 21 or 23 new samples?

**Summary and General Comments**

Reviewer #1: This is a well done study submitted by an experienced team of population geneticists. They have generated a limited number of new genome-wide sequences of Plasmodium vivax from an understudied population (CMB), but extensively compare their findings with those from other sites in the GMR. The overall picture they reveal is quite interesting, with relatively little gene flow between the CMB populations and the other P. vivax populations in the region.

Reviewer #2: Report Review PNTD-D-20-00486

[April 22, 2020]

Brashear et al. present a comprehensive study on the genomic characterization of the P. vivax (Pv) population in a specific area of the China-Myanmar border (CMB) in comparison with Pv populations in the Greater Mekong Subregion (GMS) countries Thailand, Cambodia, Vietnam and Laos.

The manuscript is well written: Rationale and aims are clearly outlined in the introduction, methods are all-inclusive and state-of-the-art, and the results are presented in detail and in a logical manner. However, I have some major comments regarding their discussion of their findings and resulting conclusions.

1) The samples have been collected during a Pv outbreak in this region in 2013. Hence, overall results do not come as a surprise to me. Instead of concluding an outbreak from their data, I would have expected to find this information in the ‘introduction’ or the ‘’methods’ and data be interpreted in view the fact that these are – indeed – samples from a single outbreak. This is, a limitation of the study, particularly in their comparative analyses of the CMB population with the other GSM populations: If infections emerge from a clonal expansion in one population, this can significantly skew further downstream analyses comparing different populations. I suggest to re-analyse the data under different threshold scenarios, with the threshold referring to how independent=unrelated single samples of the CMB population are. Depending on the threshold, (i.e., how many same or highly related samples are removed from the CMB population) your overall population size may shrink, but the remaining samples may be more representative of the CMB population.

2) The authors claim their 36 SNP panel to be an ideal set for this region for monitoring the gene flow across borders. However, for the very same reason (i.e., clonal expansion), this SNP set my not be able to be used in other regions, even not along the entire CMB. I acknowledge that the authors state that the SNP set has to be validated further in the whole of the GMS. However, I’d prefer the authors to be a bit more cautious with this conclusion, particularly in view of the fact that the Thai data set comes from the Thai-Myanmar border area where significant gene flow across borders has been shown.

More minor comments below:

General

3) The authors should write in full an abbreviation the first time it occurs in the manuscript.

4) There seems to be a ‘glitch’ in the reference managing program.

Methods

5) Data availability: Data are not available in the NCBI database under bioproject PRJNA603279. Will they be uploaded upon publication? I suggest uploading to the European Nucleotide Archive (ENA).

6) Lines 147-148: The first sentence can be deleted. Stated in lines 133-134.

Results

7) Table titles should appear on top of the table and footnotes at the bottom.

8) Authors compared their regional 36 SNP barcode with the Broad Institute global 42 SNP barcode and the LSTMH 72 SNP barcode. It would be interesting to see how it compares to the recently uploaded 65 SNP 9 (full) and 28 SNP (core) barcodes [https://www.biorxiv.org/content/10.1101/776781v1].

9) Line 366: “services” should read “surfaces”.

10) Lines 426 and 435: REF #38, #49, and #40 can be omitted.

11) Line 435: “pfmdr1” should read “pvmdr1. Please note: The role of pvmdr1 in mediating chloroquine resistance in P. vivax is still controversial; the authors should consider this.

12) Figure 3C: Data could be visualized better in a rooted tree.

Discussion

See comments #1 and #2.

13) Line 515: Change to “…and used this information to characterize the P. vivax population in this region.”

14) Line 557: Ref #36, #43, and #44 should be removed.

15) Line 609: Change to “…emerging chloroquine resistance in the northeaster Myanmar P. vivax population”.

Reviewer #3: This study examined the genetic relatedness and genomic features of P. vivax including 21 new genomes from the CMB and 200+ published genomes from other parts of the Great Mekong Subregion. The authors found clear differentiation between the CMB and eastern GMS samples. In particularly, they identified a panel of 36 high-performance SNPs that can provide country-level resolution. They examined further selection genome-wise and identified a number of resistance genes that have undergone different level of fixation among regions. I applaud the authors doing a thorough analyses on the vivax genomes in the region. The manuscript is very well written and easy to follow. Results corroborate some of the previous findings. I have a few minor comments and suggestions.

PLOS authors have the option to publish the peer review history of their article (what does this mean?). If published, this will include your full peer review and any attached files.

Reviewer #1: Yes: Marcelo Urbano Ferreira

Reviewer #2: No

Reviewer #3: Yes: Eugenia Lo
---

## [Decision Letter · Decision Letter 1]

15 Jun 2020

Dear Dr. Cui,

Thank you very much for submitting your manuscript "Population genomics identifies a distinct Plasmodium vivax population on the China-Myanmar border of Southeast Asia" for consideration at PLOS Neglected Tropical Diseases. As with all papers reviewed by the journal, your manuscript was reviewed by members of the editorial board and by several independent reviewers. The reviewers appreciated the attention to an important topic. Based on the reviews, we are likely to accept this manuscript for publication, providing that you modify the manuscript according to the review recommendations. 

Sincerely,

Kamala Thriemer

Guest Editor

Hans-Peter Fuehrer

Deputy Editor

Reviewer's Responses to Questions

**Key Review Criteria Required for Acceptance?**

**Methods**

-Are the objectives of the study clearly articulated with a clear testable hypothesis stated?

-Is the study design appropriate to address the stated objectives?

-Is the population clearly described and appropriate for the hypothesis being tested?

-Is the sample size sufficient to ensure adequate power to address the hypothesis being tested?

-Were correct statistical analysis used to support conclusions?

-Are there concerns about ethical or regulatory requirements being met?

Reviewer #2: Methods are all-inclusive, appropriate, and state-of-the-art.

**Results**

-Does the analysis presented match the analysis plan?

-Are the results clearly and completely presented?

-Are the figures (Tables, Images) of sufficient quality for clarity?

Reviewer #2: Results are presented clearly in great detail and in a logical manner.

**Conclusions**

-Are the conclusions supported by the data presented?

-Are the limitations of analysis clearly described?

-Do the authors discuss how these data can be helpful to advance our understanding of the topic under study?

-Is public health relevance addressed?

Reviewer #2: The discussion is appropriate and all-inclusive. However, I miss the discussion of the limitations of their study. See 'general comments' below.

**Editorial and Data Presentation Modifications?**

Reviewer #2: (No Response)

**Summary and General Comments**

Reviewer #2: The authors adequately addressed all comments of the three Reviewers and modified the manuscript accordingly. Apart from minor editorial changes that are required (i.e., typos, etc.), there a couple of additional comments I would like to add:

The authors compare their 36-NP panel/barcode with the global

a) 42-SNP barcode (Neafsey et al., 2012),

b) 71-SNP barcode (Diez Benavente et al., 2020), and

c) 65-SNP (full) and 28-SNP (core) barcodes (Trimarsanto et al., bioRxiv 2020).

1) Apparently, not all SNPs of the alternative barcodes were callable.

What was the reason for this?

2) They claim that the Matthew correlation coefficient (MCC) scores were lower in these alternative barcodes than in their 36-SNP barcode. However, there is no mention in the 'Methods" about how they calculated the MCC scores and more importantly, they do not give the MCC score of their 36-SNP barcode.

I am confident that these minor revisions can be made at the discretion of the authors.

PLOS authors have the option to publish the peer review history of their article (what does this mean?). If published, this will include your full peer review and any attached files.

Reviewer #2: Yes: Jutta Marfurt
---

## [Editor Report · Decision Letter 2]

22 Jun 2020

Dear Dr. Cui,

We are pleased to inform you that your manuscript 'Population genomics identifies a distinct Plasmodium vivax population on the China-Myanmar border of Southeast Asia' has been provisionally accepted for publication in PLOS Neglected Tropical Diseases.

Best regards,

Kamala Thriemer

Guest Editor

Hans-Peter Fuehrer

Deputy Editor

---

## [Editor Report · Acceptance letter]

27 Jul 2020

Dear Dr. Cui,

We are delighted to inform you that your manuscript, "Population genomics identifies a distinct Plasmodium vivax population on the China-Myanmar border of Southeast Asia," has been formally accepted for publication in PLOS Neglected Tropical Diseases.

Best regards,

Shaden Kamhawi

co-Editor-in-Chief

Paul Brindley

co-Editor-in-Chief
